# Induced Mitochondrial Alteration and DNA Damage via IFNGR-JAK2-STAT1-PARP1 Pathway Facilitates Viral Hepatitis Associated Hepatocellular Carcinoma Aggressiveness and Stemness

**DOI:** 10.3390/cancers13112755

**Published:** 2021-06-02

**Authors:** Yih-Giun Cherng, Yi Cheng Chu, Vijesh Kumar Yadav, Ting-Yi Huang, Ming-Shou Hsieh, Kwai-Fong Lee, Wei-Hwa Lee, Chi-Tai Yeh, Jiann Ruey Ong

**Affiliations:** 1Department of Anesthesiology, School of Medicine, College of Medicine, Taipei Medical University, Taipei 110, Taiwan; stainless@s.tmu.edu.tw; 2Department of Anesthesiology, Taipei Medical University-Shuang Ho Hospital, New Taipei 23561, Taiwan; 3Department of Medicine, St. George’s University School of Medicine, St. George SW17 0RE, Grenada; ychu@sgu.edu; 4Department of Medical Research & Education, Taipei Medical University-Shuang Ho Hospital, New Taipei 235, Taiwan; 20604@s.tmu.edu.tw (V.K.Y.); 15729@s.tmu.edu.tw (T.-Y.H.); 20213@s.tmu.edu.tw (M.-S.H.); 5Biobank Management Center, Taipei Medical University-Shuang Ho Hospital, New Taipei 23561, Taiwan; 19118@s.tmu.edu.tw; 6Department of Pathology, Taipei Medical University-Shuang Ho Hospital, New Taipei 23561, Taiwan; whlpath97616@s.tmu.edu.tw; 7Department of Medical Laboratory Science and Biotechnology, Yuanpei University of Medical Technology, Hsinchu 300, Taiwan; 8Department of Emergency Medicine, Taipei Medical University-Shuang Ho Hospital, New Taipei 23561, Taiwan; 9Graduate Institute of Injury Prevention and Control, Taipei Medical University, Taipei 110, Taiwan; 10Department of Emergency Medicine, School of Medicine, Taipei Medical University, Taipei 110, Taiwan

**Keywords:** hepatocellular carcinoma, IFNGR-JAK-STAT-PARP1 axis pathway, momelotinib, EMT, stemness

## Abstract

**Simple Summary:**

Hepatitis virus is a major risk factor for liver cancer. We analyzed possible synergism between momelotinib and sorafenib in hepatitis virus-associated liver cancer. The combined effect of momelotinib and sorafenib both at in vitro and in vivo synergistically sup-presses the proliferation of vHCC cells and effectively reduces the tumor burden. Our results showed that momelotinib effectively suppressed the expression of the IFNGR-JAK-STAT-PARP1 pathway, which results in the downregulation of cancer stem cell genes and enhances the antitumor efficacy of sorafenib by initiating the expression of apoptosis-related genes and inhibiting the DNA repair gene in vHCC cells, thus maximizing its therapeutic potential for patients with HCC.

**Abstract:**

Background: Hepatitis virus is a major risk factor for liver cancer. The mitochondrial dysfunction IFN gamma-related pathways are activated after virus infection. Jak family-related protein is involved in the downstream of IFN gamma-related pathways. However, the effect of the IFNGR-JAK-STAT pathway acting as functional regulators of their related protein expression on virus infection and hepatocellular carcinoma (HCC) remains unclear. Interestingly, the role of the DNA repair gene (PARP1) in therapy resistant cancers also has not been studied and explored well. In this study, we hypothesized that momelotinib could suppress the progression of HCC by targeting Jak family related and PARP1 DNA repair protein. Based on this observation, we link the relevant targets of the JAK family and the potential applications of targeted therapy inhibitors. Methods: We analyzed possible synergism between momelotinib and sorafenib in hepatitis virus-associated liver cancer. Immunostaining, colony formation assay, cell invasion, migration, and tumorsphere-formation assay were used for drug cytotoxicity, cell viability, and possible molecular mechanism. Result: We first demonstrated that the expression of Jak1 and 2 is significantly upregulated in vHCC than in nvHCC/normal liver tissues. In addition, the gene expression of IFN gamma-related pathways is activated after virus infection. Additionally, we found that momelotinib significantly inhibited the growth of HCC cells and reduces the expression of Jak2, which showed the importance of momelotinib in targeting Jak2 and reducing tumorigenesis in HCC. Meanwhile, momelotinib effectively inhibited the IFNGR-JAK-STAT pathway and reduced the migratory/invasive ability of vHCC cells through down-regulating EMT biomarkers (E-cadherin and vimentin), transcription factor (Slug), and significantly inhibits the DNA damage repair enzyme PARP1. It also induced cell apoptosis of vHCC cells. Furthermore, the combined effect of momelotinib and sorafenib both at in vitro and in vivo synergistically suppresses the proliferation of vHCC cells and effectively reduces the tumor burden. Conclusions: Our results showed that momelotinib effectively suppressed the expression of the IFNGR-JAK-STAT-PARP1 pathway, which results in the downregulation of cancer stem cell genes and enhances the antitumor efficacy of sorafenib by initiating the expression of apoptosis-related genes and inhibiting the DNA repair gene in vHCC cells, thus maximizing its therapeutic potential for patients with HCC.

## 1. Introduction

Hepatocellular carcinoma (HCC) is one of the most common cancer-associated diseases worldwide [1], and it accounts for more than 80% of primary liver cancers [2]. HCC poses a heavy disease burden, and the total number of liver cancer cases has been increasing with aging [2]. HCC is associated with liver cirrhosis in approximately 80–90% of cases [3]; patients with chronic viral hepatitis B and C infection have a high risk of HCC [4]. The risk of HCC increases after the development of cirrhosis, increasing the likelihood of death due to liver failure or HCC [5]. More than half a million individuals worldwide are annually diagnosed with HCC [6]. Currently, there are limited treatment options for HCC. For the past two decades, the median survival time for patients with advanced HCC is less than one year, and the 5-year relative survival rate is below 9% [7]. Patients with well-preserved and good liver function typically opt for surgical resection of the liver. However, the most effective treatment method to improve the survival of patients with HCC is liver transplantation [8]. Unfortunately, this treatment often results in a poor prognosis, including a high risk of postoperative complications and tumor recurrence. Several strategies are available to extend the survival time of patients with liver cancer, such as transplantation; surgical resection; target drugs, such as sorafenib, lenvatinib, and regorafenib; and immunotherapy (nivolumab) [9].

Janus kinases (Jaks) are nonreceptor tyrosine kinases encoded by the Jak gene. The Jaks family comprises four members, namely Jak1, Jak2, Jak3, and Tyk2. The Jak2 gene is a major component of the Jak-Stat signaling pathway, which is linked by interferon-responsive genes to signal transduction [10,11]. Jak2 was also found to participate in several different types of cancers and myeloproliferative diseases [12,13,14,15]. Currently, several important Jak2 inhibitors, such as ruxolitinib, pacritinib, fedratinib, and momelotinib, are available, and they differ from each other with respect to the inhibition of other kinases [16]. Interestingly, in vitro treatment with a combination of chemotherapy and momelotinib is a potent inhibitor of Jak2; it suppressed CSCs-like cells and reduced tumor burden in a mouse model of human ovarian cancer [17]. Epidemiological studies report that the hepatitis virus B and C (HBV and HCV) often were considered as the most common etiology of HCC, and these two viruses together were often known as one risk factor for HCC [18]. Viral infection-associated HCC (vHCC) often develops resistance to sorafenib (a multi-kinase inhibitor) therapy [19]. Thus, an important target that targets vHCC and overcomes drug resistance is required. Poly(ADP-ribose)polymerase (PARP)1 is an important member of the PARP family [20], which upon activation promotes DNA repair in eukaryotes1. However, much evidence suggests PARP1 can go beyond DNA repair and may regulate many other key biological processes, such as chromatin strengthening, stress signaling, cell survival/death, inflammation, drug resistance, and differentiation. PARP1 also functions as a sensor of oxidative stress, and PARP1 inhibitors have been seen enhancing the cancer cell’s sensitivity towards chemotherapy. Many cell-based studies denoted that PARP1 interacts directly with STAT3 and mediated in vitro poly(ADP-ribosyl)action of STAT3 [21]. However, the role of PARP1 linked with JAK-STAT signaling in drug resistant liver cancer has not been explored well.

In this study, we hypothesized that momelotinib could regulate the progression of HCC by targeting Jak2 and associated markers. Our study suggests that Jak2 and PAPR1 are expressed higher in vHCC than in non-viral infection-associated HCC (nvHCC) and the normal liver. Jak2 expression was correlated with poor prognosis and poor overall survival of patients with HCC. Treatment of momelotinib significantly inhibits Jak2, resulting in the reduction of the migratory, invasive property of vHCC cells. Interestingly, cell cycle arrest and inhibition of the stem cell-like phenotype of vHCC cells were also observed after the momelotinib treatment. Furthermore, the combined effect of momelotinib and sorafenib both at in vitro and in vivo synergistically suppresses the proliferation and targets the DNA repair enzyme of vHCC cells, and effectively reduces the tumor burden and therapy resistance.

## 2. Materials and Methods

### 2.1. Ethics Approval and Consent to Participate

We obtained 30 matched nvHCC, and vHCC samples as a kind gift from Dr. Wei-Hwa Lee, from the Department of Pathology, Taipei Medical University-Shuang Ho Hospital HCC tissue bank, following ethical approval for their use from the Institutional Review Board of the Taipei Medical University-Shuang Ho Hospital. The requirement for patients to sign informed consent was waived because tissue samples were obtained retrospectively from the Taipei Medical University-Shuang Ho Hospital HCC archive. This study was conducted in a cohort of patients with HCC cancer at Taipei Medical University Shuang-Ho Hospital, Taipei, Taiwan. The study was reviewed and approved by the institutional review board (TMU-JIRB: 201302016).

### 2.2. Cell Lines and Reagents

The human SNU-387 and SNU-475 HCC cell lines were purchased from American Tissue Culture Collection (ATCC). The cells were maintained under conditions recommended by the vendor, cultured in Roswell Park Memorial Institute (RPMI) 1640 Medium (Thermo Fisher Scientific Inc., Waltham, MA, USA.) supplemented with 10% fetal bo-vine serum (FBS) and 1% penicillin-streptomycin (Invitrogen, Life Technologies, Saint-Louis, MO, USA) at 37 °C, in a 5% humidified CO_2_ incubator. THLE-2 cell line was derived from primary normal liver cells by infection with SV40 large T antigen, HepG_2_ was derived from a liver hepatocellular carcinoma both were purchased from ATCC. THLE-2 and HepG_2_ cell lines were maintained in DMEM medium (Dulbecco’s Modified Eagle Medium; Gibco, Invitrogen), and all cells were incubated at 37 °C in a humidified incubator containing 5% CO_2_ in air. Cells were sub-cultured at 80–90% confluency. The Jak inhibitor momelotinib (CYT387) [21] and sorafenib (Catalog No. S7397), purchased from Selleck Chemicals, were dissolved in DMSO.

### 2.3. Microarray and RNAseq Pre-Processing and Analysis

Gene expression profiles, GSE14323, GSE14520, and GSE6764, were downloaded from the Gene Expression Omnibus (GEO) database (www.ncbi.nlm.nih.gov/geo/, accessed on 4 May 2020), and the data of RNA-seq expression results containing samples of vHCC and nvHCC from patients with liver hepatocellular carcinoma (LIHC) were downloaded from TCGA portal Xena browser (https://xenabrowser.net/, accessed on 10 May 2020) used for survival analysis. GEO dataset GSE62813 was used to identify sorafenib resistance in HCC.

### 2.4. Differential Expression Analysis

To determine differentially expressed genes between tumor and adjacent normal liver tissues in the GSE14520 cohort, the Bayes method and a linear model were employed using the R package limma. *p* values were adjusted using the false discovery rate controlling the Benjamini–Hochberg procedure. Genes with log2 fold change >1 and adjusted *p* values < 0.05 were considered significant.

### 2.5. Human Specimens and Immunohistochemistry (IHC)

Tumor tissues were harvested from patients with HCC. Samples were collected from the Department of Pathology, Shuang Ho Hospital, after obtaining written informed consent from patients (TMU-JIRB: 201302016). Clinical details of patients from the Taipei Medical University Shuang-Ho Hospital HCC cancer cohort shown in Appendix A. Using virus infection as a distinguishing criterion, Appendix A contains 30 samples, which are divided into three groups, including normal tissues (10 samples), non-viral hepatocellular carcinoma (10 samples), and viral hepatocellular carcinoma (10 samples). The collected samples were fixed in 4% paraformaldehyde and embedded in paraffin, with 5-µm sections cut from paraffin blocks. Staining was performed using anti-Jak2 (1:500; cat. ab39636, Abcam), followed by staining using the secondary antibody and H&E. The percentage of stained area to the selected field was recorded in a 5% interval, ranging from 0% to 100%. The staining intensity was graded into 3 categories (absent or weak, 1; moderate, 2; strong, 3). Q-score was derived from the product of percentage (P) of tumor cells with characteristic IHC staining (0–100%) and the intensity (I) of IHC staining (1-3) (Q = P × I; maximum = 300).

### 2.6. Cell Viability Test and Calculation of the Combination Index

The stock of momelotinib and sorafenib was prepared by dissolving 20 mg/mL of the mixture in DMSO. The stocks of each drug were stored at −20 °C until use. Using the CompuSyn software to calculate the half-maximal inhibitory concentration (IC_50_) values of different cell lines as previously reported by Chou TC and Martin N. The calculation method of IC_50_ is as described in the PC Software and User’s Guide on the ComboSyn Inc. website (http://www.combosyn.com, accessed on 4 May 2020). The effects of momelotinib and sorafenib on cell proliferation were detected using the sulforhodamine B (SRB) assay. The synergistic effect of these two drugs was analyzed using isobolograms of the drug combination, as previously reported by Chou and Talalay [22,23]. The interaction between the two drugs was also analyzed by the median-effect principle proposed by them. The combination index (CI) was calculated using CompuSyn software. A CI value is less than 1 represented synergism [23]. Briefly, HCC cells or tumorspheres were seeded in 96-well plates (8 × 10^5^ cells/well) and treated with the drugs (momelotinib or sorafenib alone or in combination) at the 5 μM and 2.5 μM concentration for 48 h, respectively. After respective drug treatments, the relative cell number was estimated by the SRB reagent according to the manufacturer’s protocol (Sigma, Vallejo, CA, USA).

### 2.7. Transient Transfection and Dual-Luciferase Assay

SNU-387 and SNU-475 cells were seeded at a density of 1 × 10^6^ cells in 100-mm2 culture plates. On the following day, cells were treated with TransFectin (Bio-Rad Laboratories, CA, USA) and transfected with 2 µg of pJAK2-TA-Luc and 2 µg of pRL-TK (Renilla luciferase control reporter plasmid (Promega, Foster, CA, USA). After 5 h of transfection, cells were trypsinized and seeded onto sterile, black-bottomed 96-well plates at a density of 1 × 10^4^ cells per well and then incubated with the complete medium for 24 h. Cells were treated with either test compounds or 0.1% DMSO for 24 h. After treatment, cells were harvested in 20 µL of passive lysis buffer, and luciferase activity was evaluated using the dual-luciferase reporter assay kit (Promega) on Wallac 1420 VICTOR2 (PerkinElmer, Inc., Los Angeles, CA, USA). The experiments were performed in triplicate and repeated thrice. Relative luciferase activity was calculated according to the following formula: relative luciferase activity (%) = ([normalized luciferase activity of sample treated with a test compound]/[normalized luciferase activity of sample treated with 0.1% DMSO]) × 100.

### 2.8. Apoptosis and Cell Cycle Analysis

Apoptosis and cell cycle were analyzed through flow cytometry (Beckman, Fullerton, CA, USA) using the Annexin V/7AAD (FITC-conjugated) apoptosis kit (F-6012; US Everbright Inc., San Ramon, CA, USA) or propidium iodide (PI) according to the manufacturer’s protocol. To determine the effect on the cell cycle, HCC cells were exposed to momelotinib for 48 h. Thereafter, cells were washed and fixed with 70% ethanol. The cells were washed, re-suspended, and stained with 10 µg/mL of PI in PBS for 30 min at room temperature in the dark. The cells were analyzed through flow cytometry (Becton Dickinson, Mountain View, CA, USA), and the population of cells in each phase was counted.

### 2.9. TUNEL Assay

Cell death (apoptosis) in the tumor tissue was detected by staining HCC tissues using an in situ cell death detection kit from Roche according to the manufacturer’s protocol. Quantification was performed by calculating the percentage of TUNEL-positive cells by using a fluorescence microscope. The results are expressed as the mean number of TUNEL-positive apoptotic HCC cells in each group.

### 2.10. ALDEFLUOR Assay and ALDH1+ Cell Sorting by FACS

ALDH activity in HCC cells was assayed using the ALDEFLUOR kit according to the manufacturer’s instructions (STEMCELL Technologies, Durham, NC, USA). The cells that showed positive activity of aldehyde dehydrogenase 1 (ALDH1) were isolated and analyzed. Briefly, the human HCC cell lines SNU-387 was suspended at a concentration of 1 × 10^6^ cells/mL in ALDEFLUOR assay buffer containing the ALDH substrate (BAAA, 1 μmol/L per 1 × 10^6^ cells) and incubated for 40 min at 37 °C. Cells incubated with ALDEFLUOR substrate and treated with 50 mmol/L of diethylaminobenzaldehyde (DEAB), a specific ALDH inhibitor, were used as a reference control. To avoid contamination of cells of mouse origin from xenotransplanted tumors, they were stained with the anti-H2Kd antibody (BD Biosciences, 1/200, 20 min on ice) and then with a secondary antibody labeled with phycoerythrin (Jackson Labs, 1/250, 20 min on ice). Cells stained with PI alone were used as negative controls, and ALDEFLUOR-stained cells treated with DEAB and those stained with the secondary antibody alone were considered viable.

### 2.11. Western Blotting and qRT-PCR

Cells were washed with PBS and then lysed in RIPA lysis buffer; cellular protein lysates were isolated using the protein extraction kit (QIAGEN, Portland, OR, USA) and quantified using the Bradford protein assay kit (Bio-Rad, Taiwan). Approximately 20 μg of the sample from different experiments were loaded and subjected to SDS-PAGE by using the Mini-PROTEAN III system (Bio-Rad, Taipei, Taiwan). Separated proteins were transferred onto a polyvinylidene fluoride (PVDF) membrane by using the Trans-Blot Turbo Transfer System (Bio-Rad), followed by blocking with Tris-buffered saline plus skimmed milk. These PVDF membranes were then probed with respective primary antibodies, followed by the secondary antibody. The commercial antibodies are shown in Appendix A. The enhanced chemiluminescence detection kit was used to detect the proteins of interest. Images were captured and analyzed using the UVP BioDoc-It system (Upland, CA, USA). qRT-PCR was performed by using isolating total RNA using TRIzol-based protocol (Life Technologies, San Jose, CA, USA) provided by the manufacturer. In brief, one µg of total RNA was reverse transcribed using QIAGEN OneStep RT-PCR kit (QIAGEN, Taipei, Taiwan), and PCR was performed using a Rotor-Gene SYBR Green PCR kit (QIAGEN, Tiapei, Taiwan). The primer sequences are shown in Appendix A.

### 2.12. Colony Formation Assay

The colony formation assay was performed according to a previously explained protocol [24] with modifications. Briefly, a total of 500 colon cancer cells were seeded in six-well plates and treated with momelotinib. The cells were allowed to grow for another week and then harvested, fixed, and counted.

### 2.13. Wound Healing Migration Assay

Cells were seeded in six-well plates (Corning, New York, NY, USA) with RPMI 1640 medium containing 10% FBS and cultured to 95–100% confluence. A scratch along the median axis was then made with a sterile yellow pipette tip across cells. Cell migration pictures were captured at 0 and 48 h after the medium scratch, under a microscope, and analyzed using the NIH Image J software. (https://imagej.nih.gov/ij/download.html, accessed on 4 May 2020).

### 2.14. Matrigel Invasion Assay

Cells (2 × 10^5^) were seeded in 24-transwell chambers with an 8-μm pore membrane coated with Matrigel in the upper chamber of the transwell system containing serum-free RPMI 1640 medium. The lower chamber of the transwell system contained the medium with 20% FBS. After incubation at 37 °C for 6 h, non-invaded HCC cells on the upper side of the membrane were carefully removed with a cotton swab, and the invaded cells were stained with crystal violet dye, air-dried, and photographed under a microscope. Images were analyzed using the NIH Image J software.

### 2.15. Sphere Formation Assay

Cells (5 × 10^3^/well) were plated in ultra-low-attachment six-well plates (Corning) containing stem cell medium consisting of serum-free RPMI 1640 medium supplemented with 10 ng/mL of human basic fibroblast growth factor (bFGF; Invitrogen, Grand Island, NY, USA), 1 × B27 supplement, and 20 ng/mL epidermal growth factor (EGF; Invitrogen). The medium was changed every 72 h. After 14 days of incubation, the formed spheres were counted and photographed.

### 2.16. Animal Studies

All animal experiments and maintenance were in strict compliance with the Animal Use Protocol Taipei Medical University (protocol LAC-2019-0526). The anti-proliferative effect of momelotinib and sorafenib in combination with vHCC cells in vivo investigated, athymic nude mouse models bearing HCC cell xenografts were established. Five-week-old male athymic nude mice were used for this study. The animal experiment is set to six mice per group. The mice were maintained under pathogen-free conditions and were provided with sterilized food and water. First, 1 × 10^6^ SNU-387 cells were subcutaneously injected into the right flank near the hind leg of each nude mouse. When the mice had palpable tumors (tumor volume of approximately 100 mm^3^), they were randomly divided into control (100 µL of normal saline [NS] by intraperitoneal injection plus 100 µL of 1% DMSO) and 0.5% carboxymethyl cellulose ([CMC]-Na sterile water), momelotinib (30 mg/kg/day by intraperitoneal injection plus 100 µL of 1% DMSO and 0.5% CMC-Na sterile water), sorafenib (30 mg/kg/day by intragastric administration plus 100 µL NS by intraperitoneal injection), and combination (momelotinib, 30 mg/kg/day by intraperitoneal injection plus sorafenib 30 mg/kg/day by intragastric administration) groups (n = 6 animals/group). The treatments were performed 5 times/week for 4 weeks. The tumor volume was detected every week and was calculated using the following formula: volume (V) = π/6 × length × width × height. After 4 weeks, mice were humanely euthanized, and the tumors were isolated for further analyses.

### 2.17. Statistical Analysis

All assays were performed at least thrice in triplicate. Values are expressed as the mean ± standard deviation (SD). Comparisons between groups were estimated using Student’s *t*-test for cell line experiments or the Mann–Whitney U-test for clinical data, Spearman’s rank correlation between variables, and the Kruskal–Wallis test for comparison of three or more groups. The Kaplan–Meier method was used for the survival analysis, and the difference between survival curves was tested by a log-rank test. Univariate and multivariate analyses were based on the Cox proportional hazards regression model. All statistical analyses were performed using IBM SPSS Statistics for Windows, version 20 (IBM, Armonk, NY, USA). A *p*-value < 0.05 was considered statistically significant.

## 3. Results

### 3.1. IFNGR-JAK-STAT1 Signaling Pathway Is Activated in vHCC

We investigated the expression of IFNGR-JAK-STAT in hepatitis C virus (HCV)-associated HCC (vHCC), non-HCV-associated HCC (nvHCC), and normal liver samples. The robust microarray analysis expression data for liver samples from GEO datasets GSE14323 were explored, and 10 patients containing both normal liver and vHCC sample pairs were obtained after gene expression profiling within this sample group by using a heatmap. Interestingly, the expression of Jak2 in HCC cells was higher than that in normal liver cells (Figure 1A). The differential expression of IFNGR1, IFNGR2, Jak1, Jak2, and STAT from the entire series was analyzed; as expected, the normal cohort (n = 19) showed lower Jak2 mRNA expression than did the HCC (HCV-infected) cohort (n = 38), and the GSE121248 cohort showed the similar result that Jak1 mRNA expression is higher in HCC (HBV-infected, n = 70) than in the normal liver (n = 37; Figure 1B). Protein-protein interaction (PPI) network of IFNGR-JAK-STAT signaling proteins by STRING, clustering into three groups with blue, green, and red, representing (Figure 1C). Intriguingly, the expression of Jak2 in nvHCC groups from TCGA LIHC data remained similar compared with the normal liver group (Figure 1D), suggesting that Jak2 appears to be overexpressed only in vHCC. Figure 1E summarizes the heat map of different hepatitis virus-associated liver cancer samples and shows the connection between the JAK2, STAT1 expression, and virus infection. From the results in Figure 1E, we analyzed the overall survival of HCC patients with viral and non-viral infections. First, patients with vHCC have a poor overall survival rate. When the experiment distinguished between vHCC and HCC patients, we further discovered a higher expression of STAT1 and Jak2 is also associated with lower overall survival of patients with vHCC (Figure 1F). Further analyzing the relationship between the STAT1 gene expression and the overall survival of patients. We were surprised to find that the patients with high STAT1 expression have a lower overall survival rate in the vHCC group and it is statistically significant. In addition, this phenomenon was not found in the overall survival of HCC patients. Figure 1G shows the high correlation between JAK2, and STAT1 in TCGA (gene expression) and SHH vHCC patients (protein expression) cohort. In addition, we also performed an association analysis of gene expression (Appendix A), and the results showed that the expression of JAK2 was significantly associated with IFNGR1/2 and STAT1 (*p* < 0.001). The expression of PARP1 was significantly associated with STAT1 (*p* < 0.001). The STAT1 is a downstream gene of JAK2, a novel pathway of IFNGR-JAK2-STAT1-PARP1 is connected in series.

Verified results show that whether it is a gene or protein expression, the correlation between JAK2 and STAT1 shows similar trends in the two data sets from different sources. To further explore the importance of JAK2 in viral hepatocellular carcinoma, we observed the expression of JAK2 in patient tissues. Clinical details of HBV- and HCV-infected patients from the Taipei Medical University Shuang-Ho Hospital HCC cancer cohort shown in Appendix A. The results showed that patients related to the hepatitis virus accounted for one-half of the primary tumor. Among patients with viral hepatitis, two patients were found to be infected with HBV and HCV. In order to further understand the connection between the JAK2 expression and virus infection, we conducted follow-up IHC experiments of different liver samples. Furthermore, by applying data from the cohort of patients with Taipei Medical University Shuang-Ho Hospital HCC cancer to corroborate this finding, we consistently observed that although Jak2 immunostaining in nvHCC (n = 10) samples remain unchanged compared with normal liver samples (n = 10), vHCC (n = 10) samples showed elevated Jak2 levels (Figure 1H and Appendix A). Figure 1I showed the Q-score of JAK2 expression in Shuang Ho Hospital patients with Hepatocellular Carcinoma, JAK2 expression after vHCC infection is higher. The scoring matrix for the 30 samples is analyzed in Appendix A. These results indicate that Jak2 and downstream gene STAT1 overexpression is distinctively associated with vHCC, whereas Jak2 expression in nvHCC is similar to that in normal liver.

### 3.2. Momelotinib Inhibits Jak2 Expression, Leading to Decreased Cell Viability

We investigated the potential of the Jak inhibitor momelotinib (CYT387; Figure 2A) to inhibit Jak2. As expected, the expression of Jak2 and p-Jak2 in vHCC cell lines SNU-475 and SNU-387 was higher than that in nvHCC (HepG_2_) and normal liver (THLE-2) cell lines (Figure 2B). Then, using the CompuSyn software to calculate the IC_50_ values at 48 h for all four cell lines are as follows: SNU-475 = 6.3236, SNU-387 = 5.59423, HepG_2_ = 6.89772, and THLE-2 = 12.4946. According to this result, we set 5 μm as the subsequent experimental condition of the SNU cell line. The cell proliferation, as well as Jak2 luciferase activity of both SNU-475 and SNU-387 cell lines, significantly decreased under treatment with 5 μM momelotinib in a dose-dependent manner (Figure 2C,D). We also confirmed the protein expression profile of IFNGR/JAK2 axis under the drugs. Momelotinib inhibits the phosphorylation of Jak2 and STAT3, and the inhibitory effect of this phosphorylation may affect the subsequent molecular signal transmission of liver cancer cells (Figure 2E). These results suggest that vHCC is sensitive to the inhibition of P-Jak2 by momelotinib.

### 3.3. Momelotinib Inhibits Migration and Invasion and Triggers Cell Cycle Arrest of vHCC Cells

After 24 h, we observed that a higher concentration of momelotinib led to a significantly increased proportion of cells at the G_0_/G_1_ phase and decreased proportion of cells at the S phase (Figure 3A). The result indicated that momelotinib might inhibit cell growth and DNA replication. Subsequently, the effect of momelotinib on the migration and invasion of vHCC cells (SNU-475 and SNU-387) was investigated. According to the IC_50_ results in Figure 2C, treatment with 5 μM momelotinib for 48 h strongly inhibited the migration (Figure 3B) and invasion (Figure 3C) abilities of cells, indicating that the Jak2 inhibitor effectively reduced the mobility and invasiveness of vHCC cells when compared with their untreated control counterparts. EMT plays an important role in the invasion and metastasis of HCC cells [25]. Furthermore, the effect of momelotinib on EMT was evaluated. EMT biomarkers (E-cadherin and vimentin) and transcription factor (Slug) were determined through Western blotting. The results showed that the expression of Slug, N-cadherin, and vimentin was significantly decreased and that of E-cadherin, a transmembrane protein with a tumor-suppressive effect, was significantly increased (Figure 3D). These results suggest that momelotinib is effective in preventing the migration and invasion of vHCC cells.

### 3.4. Momelotinib Remarkably Suppresses Colony and Tumorsphere Formation of vHCC Cells

To further examine the effect of momelotinib in tumorigenesis, we assayed the colony and tumorsphere formation of the vHCC cell lines SNU-475 and SNU-387. Colony and tumorsphere formation assays are important for the identification of stemness [26,27]. Jak2 inhibition considerably suppressed tumorsphere and colony formation of cells (Figure 4A,B). Momelotinib effectively suppressed tumorsphere and colony formation of SNU-475 and SNU-387 cells. Tumorsphere and stemness markers, such as CD133, KLF4, and SOX2, were also significantly decreased both at the protein and mRNA level (Figure 4C) after treatment with momelotinib. Decreased generation of ALDH1+ cells was observed in the momelotinib-treated group than in the dimethyl sulfoxide (DMSO)-treated group (Figure 4D). These results indicate that the stem cell-like phenotype of vHCC is modulated by the inhibition of Jak2.

### 3.5. Momelotinib Treatment Increased Sorafenib Sensitivity of vHCC Cells

Momelotinib is a potent ATP-competitive inhibitor of JAK2, it has been used for targeted therapies of myeloproliferative tumors. A previous study showed that treatment with sorafenib significantly improved the survival of patients with solid tumors (Abou-Alfa et al., 2006). In our results, the combination of momelotinib and sorafenib suppressed the proliferation of SNU-387 cells (Figure 5A). In order to inhibit the effect of JAK2 pathway activation on drug resistance, we tested whether the combination of momelotinib and sorafenib can remarkably suppress vHCC cell proliferation and colony formation. We observed that the combination treatment with momelotinib and sorafenib synergistically inhibited vHCC cell proliferation and colony formation by inducing apoptosis. Figure 5B shows the momelotinib-sorafenib combination testing on SNU-387 and the effect of treatment on colony-forming capacity of SNU-387 cells and treated with the drugs (sorafenib and momelotinib in combination) at the 2.5 μM and 5 μM concentration for 48 h. We also calculate a synergy score for combined momelotinib and dorafenib treatment using synergy estimation tools (SynergyFinder, https://academic.oup.com/nar/article/48/W1/W488/5815821 accessed on 4 May 20) as shown in Appendix A. SynergyFinder extracts the data for all the possible drug combination pairs from the data and visualize the synergy map calculated from synergy scoring models. The result includes heatmap, contour plot and an interactive 3D surface which was demonstrated to prioritize synergistic drug pairs with higher efficacy and lower toxicity as top hits, providing thus an increased likelihood for their clinical success. In addition, we also analyzed the drug sensitivity of the HCC (HepG_2_) cell line and synergy score, shown in Appendix A. The results indicated that HepG_2_ cells had less obvious impacts on JAK2 inhibitor (momelotinib) compared with SNU-387. Momelotinib is a selective inhibitor of JAK1 and JAK2, we speculated that the hepatoma cell lines (SNU-387) infected by the virus might induce JAK2 expression through the IFN gamma pathway. This data might be supporting our proposed model of JAK2 expression in vHCC. According to these results, we found the optimized parameter of combination. In Figure 5C, we did the apoptosis analysis by flow cytometer for Annexin-V+ and 7-AAD stained cells. The results indicated the elevated apoptosis in the combo treatment. Determining the expression of apoptosis marker is important to understand the functions of molecular mechanism. Hence, Figure 5D showed the representative Western blot images of the apoptosis markers in SNU-387. The expression levels of apoptosis markers including p-JaK2 and Bcl-x1 were decline, the cleaved-PARP, cleaved-Caspase 7, and 9 were the rise in the combination-treated group. This result shows the therapeutic potential of oncogenic/Stemness JAK2 pathway inhibitors on viral hepatitis. As stated above, these results demonstrated the potential inhibition of vHCC cells by treatment with momelotinib–sorafenib combination.

### 3.6. IFNGR-JAK-STAT1-PARP1 Signaling Pathway Is Activated in vHCC

According to the experimental results in Figure 5D, we connected the potential relationship between PARP and JAK-STAT axis and inspired us to further observe the expression of PARP1 signaling proteins in related pathways. Protein-protein interaction (PPI) network of IFNGR-JAK-STAT1-PARP1 signaling proteins by STRING (Figure 6A). This shows the upstream and downstream relevance of genetic information transmission. Figure 6B shows transcription level of PAPR1 in different types of cancers (oncomine and TCGA) types. PARP1 expression associated with poor liver cancer patients’ overall survival (Figure 6C). Figure 6D, E shows the high correlation between IFNGR1, JAK2, STAT1 and PARP1 in TCGA (gene expression). PARP1 is an enzyme responsible for repairing damaged DNA and is also involved in epigenetic regulation. As far as we know, there is no report about targeting JAK2 to inhibit STAT1-PARP1 pathway. The results of this analysis may inspire more future applications of targeting JAK2 to achieve PARP inhibition and DNA damage related potential applications.

### 3.7. Momelotinib in Combination with Sorafenib Efficiently Suppressed Tumor-Initiating Ability in Xenograft Models

The effect of momelotinib on the inhibition of the tumor-initiating ability in xenograft models, HCC (SNU-387) cancer cells (1 × 10^6^ cells/injection) were subcutaneously injected into male athymic nude mice to establish a xenograft model. After 4 weeks of follow-up, mice were divided into four groups (vehicle control, momelotinib, sorafenib, and combination) when the tumor became palpable (approximately 100 mm^3^). The group that received combination treatment showed the best tumor growth inhibition. The tumor size in each treatment group was measured; a significantly smaller tumor size was found in the group that received combined treatment with momelotinib and sorafenib, indicating that the combination treatment suppressed tumorigenesis (Figure 7A). Consistent with in vitro data, immunohistochemistry, TUNEL analyses, and the overall survival rate (Figure 7B–D) of xenograft tumors revealed that the momelotinib and sorafenib combination effectively inhibited the expression of Ki-67, a marker for representing tumor proliferation. In order to show the relationship between Jak2 signaling and tumor progression. The IHC result indicated momelotinib (JAK2 inhibitor) could reduce the expression of JAK2 in vivo. Moreover, momelotinib facilitated sorafenib-induced apoptosis in xenograft tumors, as evaluated by the cleaved caspase-3 staining and TUNEL assay.

## 4. Discussion

The growing HCC is one of the leading causes of cancer-associated deaths worldwide. In more than 80–90% of cases, HCC is directly or indirectly associated with liver cirrhosis; chronic viral hepatitis B and C infections are mainly responsible for the development of HCC [1,2,3,4,5]. Often this viral infection-associated with the development of resistance to sorafenib (a multi-kinase inhibitor) therapy [20]. The overall 5 years survival rate of HCC is below 9% [8]. Janus kinases (Jaks) are nonreceptor tyrosine kinases encoded by the Jak gene. The Jaks family comprises four members, namely Jak1, Jak2, Jak3, and Tyk2 [22]. Jak2 has a role in several different types of cancers and myeloproliferative diseases [13,14,15,16]. Significantly increased expression of phosphorylated Jak2 and STAT3 are observed in Vhcc [23]. Patients with this disease develop resistance to sorafenib, a multi-kinase inhibitor. Currently, several important Jak2 inhibitors are available, such as ruxolitinib, pacritinib, fedratinib, and momelotinib [17], each of which has a different mode of action. Interestingly, the combination of chemotherapy and momelotinib is a potent inhibitor of Jak2, and it suppresses CSC-like cells and reduces tumor burden in vHCC [18]. Thus, finding an important target that re-sensitizes and overcomes the drug resistance of vHCC is required.

The Janus kinase signal transducer and activator of transcription (JAK-STAT) signal is essential for a variety of cellular processes including survival, differentiation and proliferation. Currently, four therapeutic JAK2 inhibitors (ruxatinib, melatinib, momolotinib and pacomatinib) have been approved or are in advanced clinical studies. Molotinib (CYT387) is a dual inhibitor of JAK1 and JAK2 [24]. The related downstream JAK1 and JAK2 in the IFNGR pathway induced by viral infection are related. Hence, we selected this dual inhibitor to evaluate its therapeutic effect. In addition, this is the first study to explore the application of Molotinib to viral liver cancer. In the present study, we first demonstrated that the expression of Jak1/2 and PAPR1 is significantly upregulated in vHCC than in nvHCC/normal liver tissues (Figure 1 and Figure 6). In addition, the proteins of IFN gamma-related pathways are activated after virus infection (Figure 1B). This result suggests the difference in signal transmission between viral liver cancer and general liver cancer. Based on this observation, we are linked to the relevant targets of the JAK family and the potential applications of targeted therapy inhibitors. To further confirm our hypothesis that the JAK family and virus infection cause cancer. We also analyzed the overall survival between vHCC and HCC. Among them, STAT1 expression in both is a significant difference. In virus infection, the performance of IFN-gamma is an important indicator. IFN-gamma binds to nearby uninfected cell membrane receptors, stimulating signaling pathways to interfere with virus replication; stimulating cells to hydrolyze pathogenic proteins to prevent cells from being infected by the same or different viruses. In addition, STAT1 is a downstream gene of JAK2 in IFN-gamma signaling pathways, which can infer the significance of JAK2 in viral infection. Previous research indicated that overall Jak2 expression is related to the overall survival in HCC patients [25]. Thus, the Jak family gene might be playing a key role in patients with HCC. Verstovsek et al. [26] demonstrated the efficacy of momelotinib as a potent inhibitor of Jak1 and Jak2 in patients with primary and secondary myelofibrosis. The inhibitory effect of momelotinib (CYT387) on JAK2 phosphorylation is consistent with previous reports [26]. Momelotinib (CYT387) is an ATP-competitive small molecule that potently inhibits JAK1/JAK2 kinases. Furthermore, we found that momelotinib significantly inhibited the growth of HCC cells (Figure 2) and inhibit the phosphorylation of Jak2 and STAT3 and reduce the expression of p-Jak2, which verified the importance of momelotinib in targeting Jak2 and reducing tumorigenesis in HCC. Often HCC patients show sorafenib therapy resistance [27]. Drug resistance is a major challenge in anticancer therapy. CSCs provide an alternative explanation for the aforementioned therapeutic challenges of several cancers [28]. This small population of cancer cells has stem cell-like features such as tumorigenicity, self-renewal, and more resistance to chemotherapeutic agents than that shown by cancer cells [29,30]. Treatment of vHCC with momelotinib reduced the expression of cancer stemness markers, such as CD133, KLF4, and SOX2, and decreased ALDH1 activity (Figure 4), suppressing tumorsphere and colony formation of HCC cells.

In the previous series of experiments, it has been preliminarily confirmed that the process of viral liver cancer may be related to the IFNGR-JAK-STAT pathway. Therefore, momelotinib was chosen as a JAK1/2 target combination therapy inhibitor to further prove our hypothesis. Momelotinib competes with JAK1/2 for ATP binding, which may result in inhibition of JAK1/2 activation, inhibition of the JAK-STAT signaling pathway, and so the induction of apoptosis and a reduction of tumor cell proliferation in JAK1/2-expressing tumor cells. Combination therapy, a treatment modality that combines two drugs for cancer therapy, has gained wide acceptance and popularity. It combines two drug targets involved in multiple pathways in cancer, utilizing different mechanisms to reduce the development of drug resistance in tumors [31]. Combinations of two therapeutic agents into a biological system may produce a synergistic effect, or, sometimes, antagonistic, or identical effects, compared with their effects when acting separately. Momelotinib, together with sorafenib, increased the sensitivity of HCC cells, resulting in decreased colony formation in cells treated with the combination, and also alleviated apoptotic markers (Annexin-V+ and 7-AAD positive cells) PARP-1, PDCD4, and Bax molecules (Figure 5). The integrity of the genome relies on cell cycle control and DNA repair to work together. PARP1 also functions as a sensor of oxidative stress, and PARP1 inhibitors have been seen enhancing the cancer cell’s sensitivity towards chemotherapy.

Since the role of PARP1 in DNA repair must be completed in the nucleus, it is extremely important to understand how to regulate the distribution of PARP1 in the cell. Furthermore, PARP1 inhibitors have become increasingly important in the field of cancer treatment. Considering the resistance of current PARP1 activity inhibitors, the use of regulating its cell distribution may also achieve the effect of inhibiting PARP1. Figure 6 shows PARP1 expression associated with poor liver cancer patients’ overall survival. Based on the results of these analyses, we have concluded that a novel path in which momelotinib induces DNA damage and sensitizes viral hepatitis associated hepatocellular carcinoma aggressiveness and stemness via IFNGR-JAK2-STAT1-PARP1 axis. We hope that this discovery can develop possible PARP1 nuclear metastasis inhibitors and test its possibility as cancer treatment drugs. We hope that through this research, we can further analyze the correlation between cell cycle control and DNA repair, and provide another type of PARP1 activation mechanism for cancer treatment strategies.

Similarly, the combination of momelotinib and sorafenib efficiently suppressed the tumor-initiating ability in xenograft models (Figure 7) and effectively inhibited the expression of Ki-67, a marker for representing tumor proliferation. Moreover, momelotinib facilitated sorafenib-induced apoptosis in xenograft tumors, as evaluated by the cleaved caspase-3 staining and TUNEL assay. Despite the fact that treatment with sorafenib is a useful therapeutic way for HCC, the survival rate of patients is still limited. This is because that HCC cells are heterogeneous cells with incongruous activation of several signaling pathways [32,33]. Combination therapy strategies have been proposed to improve the efficacy of sorefenib based treatment of HCC patients in many studies [34,35]. HCC patients treated with a combination therapy strategy resulted in more effective treatment, such as mTOR inhibitors and monoclonal antibodies. The JAK/STAT is an important pathway for cellular functions, including cell proliferation and differentiation, and it also participates in the mechanism of liver regeneration and gluconeogenesis [36]. Many types of research indicated that the JAK/STAT pathway is often deregulated in signaling in HCC and other cancer [37,38]. The common JAK inhibitors including pacritinib, cryptotanishinone, and ruxolitinib have been used for cancer and human-relevant disease studies. Recently, Justin Jit Hin Tang et al. (2020) discussed the JAK/STAT signaling in hepatocellular carcinoma and organized different JAK/STAT inhibitors used in targeting the JAK/STAT pathway for HCC treatment, including small molecule inhibitors and siRNAs [39]. In our report, the overexpression of Jak2 was suppressed by momelotinib, a Jak inhibitor, leading to a sharp reduction in the cell viability and DNA repair of vHCC cell lines. It has a wider significance, and this model may lead to a new therapeutic strategy for vHCC and non-vHCC.

## 5. Conclusions

In this research report, we try to explain this model, make the research context clearer through in vitro and in vivo experiments, and try to define the role of IFNGR-JAK-STAT-PARP1 pathway in the invasion of the hepatitis virus. In conclusion, as shown in the schematic diagram of Figure 8, we demonstrate that IFNGR-JAK-STAT-PARP1 pathway might facilitate viral hepatitis associated hepatocellular carcinoma aggressiveness and stemness with a series of experimental designs, which results in the downregulation of cancer stem cell genes and enhances the antitumor efficacy of sorafenib by initiating the expression of apoptosis-related genes in vHCC cells, thus maximizing its therapeutic potential for patients with HCC.

## Figures and Tables

**Figure 1 cancers-13-02755-f001:**
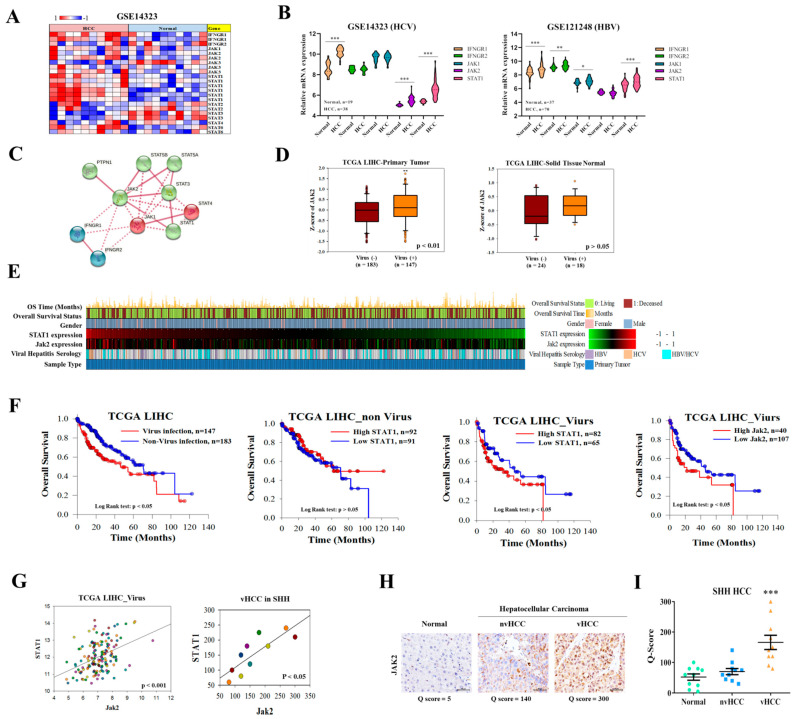
IFNGR, JAK, and STAT expression in vHCC, nvHCC, and normal liver samples. (**A**) Heatmap of 10 sample NT pairs, including both normal liver and vHCC in GSE14323 with top highly expressed genes detected by different probes. (**B**) Violin plot showed differential expression of IFNGR1, IFNGR2, Jak1, Jak2, and STAT1 in normal liver and vHCC samples from GSE14323 and GSE121248. (**C**) Schematic representation of the interaction of the IFNGR-JAK-STAT signaling proteins with the STRING analysis. (**D**) Expression of JAK2 in nvHCC and vHCC samples were analyzed from TCGA LIHC with 95% CI. (**E**) Heatmap of different Hepatitis Virus-Associated Liver cancer samples. The TCGA LIHC samples information containing overall survival status, gender, JAK2 expression, viral hepatitis serology, and sample type are shown on the left, and color block meaning is shown at the right (n = 325). (**F**) Overall survival of patients with vHCC was significantly lower than that of patients with nvHCC (*p* < 0.01, KM survival curve). (**G**) Analysis of the relationship between JAK2 and STAT1 expression in vHCC. (**H**) IHC image of Jak2 expression in normal liver, nvHCC, and vHCC samples (n = 30) from the Taipei Medical University Shuang-Ho Hospital HCC cancer cohort. (**I**) Q-score of JAK2 expression in Shuang Ho Hospital patients with Hepatocellular Carcinoma. * *p* < 0.05; ** *p* < 0.01; *** *p* < 0.001. Scale bar: 50 μm. HBV: Hepatitis B virus; HCV: Hepatitis C virus.

**Figure 2 cancers-13-02755-f002:**
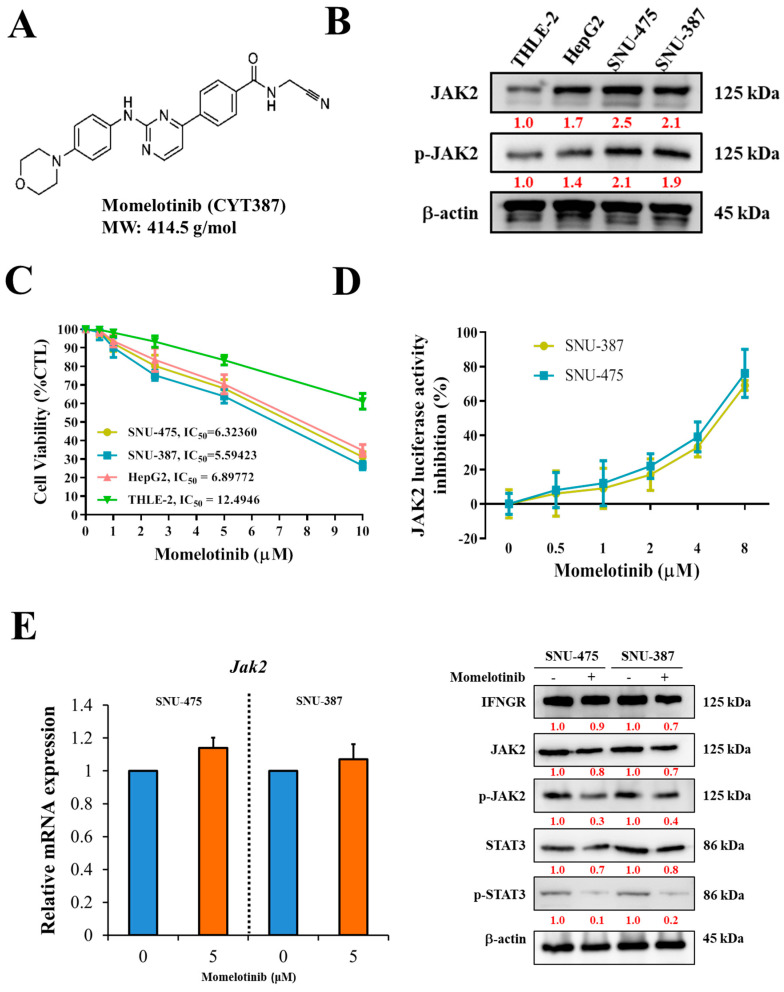
Inhibition of Jak2 by momelotinib. (**A**) Chemical structure of momelotinib (CYT387). (**B**) Immunoblot showing the expression of Jak2 and p-Jak2 in HCC cells. (**C**) Cell viability of SNU-475 and SNU-387 cell lines treated by momelotinib. (**D**) SNU-475 and SNU-387 cells were analyzed using a dual-luciferase assay. Dual-luciferase assays were performed in liver cancer cells after 24-h treatment with momelotinib. (**E**) Relative mRNA and protein expression of Jak2 after momelotinib treatment. Control (without momelotinib) groups were treated with DMSO. The momelotinib treatment dose and time with 5 μM and 48 h. β-actin was used as a loading control.

**Figure 3 cancers-13-02755-f003:**
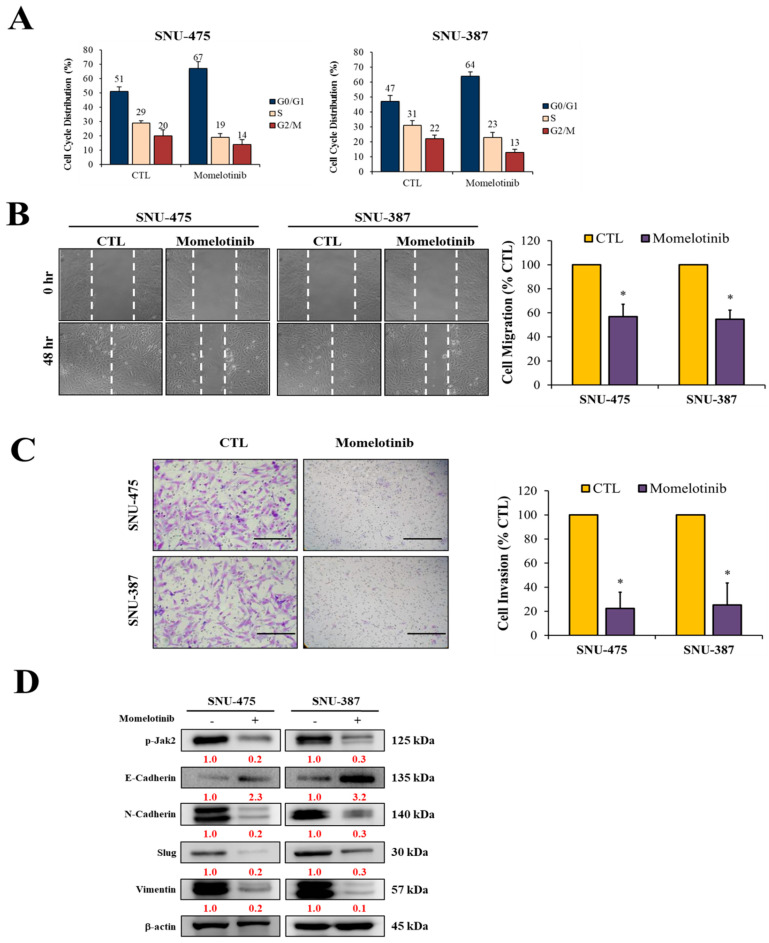
Momelotinib inhibits the migration, invasion, and induces cell cycle arrest of vHCC cells. Representative images of (**A**) momelotinib-induced cell cycle arrest, (**B**) migration, and (**C**) invasion of SNU-457 and SNU-387 cell lines. (**D**) Expression of EMT markers under treatment with momelotinib. The momelotinib treatment dose and time with 5 μM and 48h. β-actin was used as a loading control. * *p* < 0.05. Scale bar: 50 μm.

**Figure 4 cancers-13-02755-f004:**
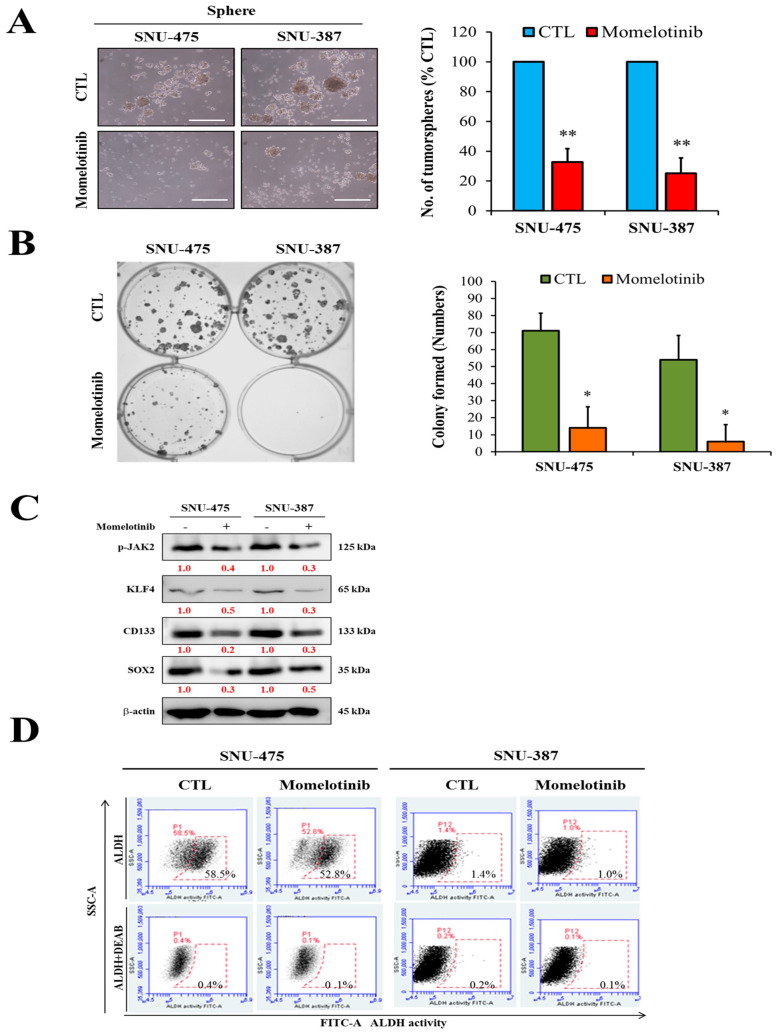
Momelotinib suppresses the stem cell-like phenotype of vHCC. (**A**) Tumorsphere-forming ability of SNU-475 and SNU-387 cells and graphical quantitation compared with their untreated control counterparts. (**B**) The colony-forming ability of SNU-475 and SNU-387 cells and quantitation compared with their treated and untreated control counterparts. (**C**) Protein and mRNA expression of stemness markers under treatment with momelotinib. β-actin was used as a loading control. (**D**) ALDEFLUOR assay showed that momelotinib treatment prominently reduces the activity of ALDH1 in SNU-475 and SNU-387 cells in a dose-dependent manner. The momelotinib treatment dose and time with 5 μM and 48 h. * *p* < 0.05; ** *p* < 0.01. Scale bar: 50 μm.

**Figure 5 cancers-13-02755-f005:**
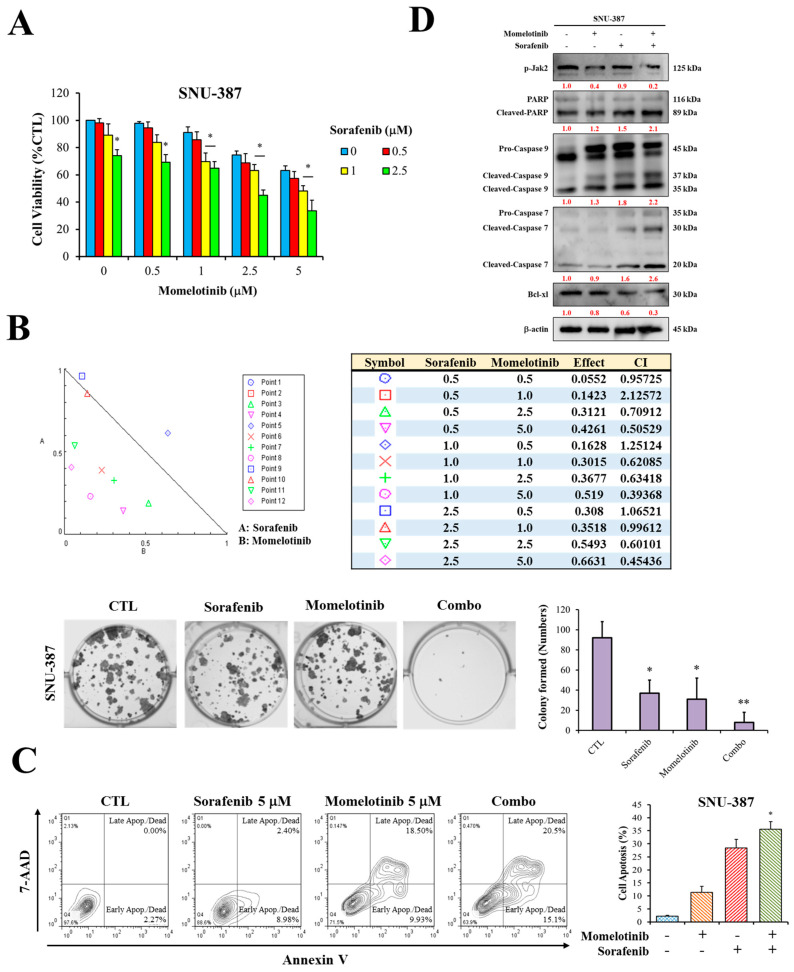
The combination of momelotinib and sorafenib suppressed the proliferation of vHCC cells. (**A**) Combination dose-dependent inhibition of vHCC cell line SNU-387 by sorafenib and momelotinib. (**B**) Synergistic effects of treatment with momelotinib–sorafenib combination on SNU-387 cells, and the effect of treatment on the colony-forming capacity of SNU-387 cells. Combination propensity and effect were evaluated using CompuSyn based on the Chou–Talalay algorithm for drug combination. Combinational effects are presented as the combination index (CI), where a CI of <1 (inside the triangle) indicates synergism, a CI of 1 (on the hypotenuse) indicates an additive effect, and a CI of >1 (outside the triangle) indicates antagonism. (**C**) Apoptosis analysis by flow cytometry for Annexin-V+ and 7-AAD-stained cells, indicating elevated apoptosis in cells treated with the combination. (**D**) Representative western blot images of apoptosis markers in SNU-387 cells. β-actin was used as a loading control. * *p* < 0.05; ** *p* < 0.01.

**Figure 6 cancers-13-02755-f006:**
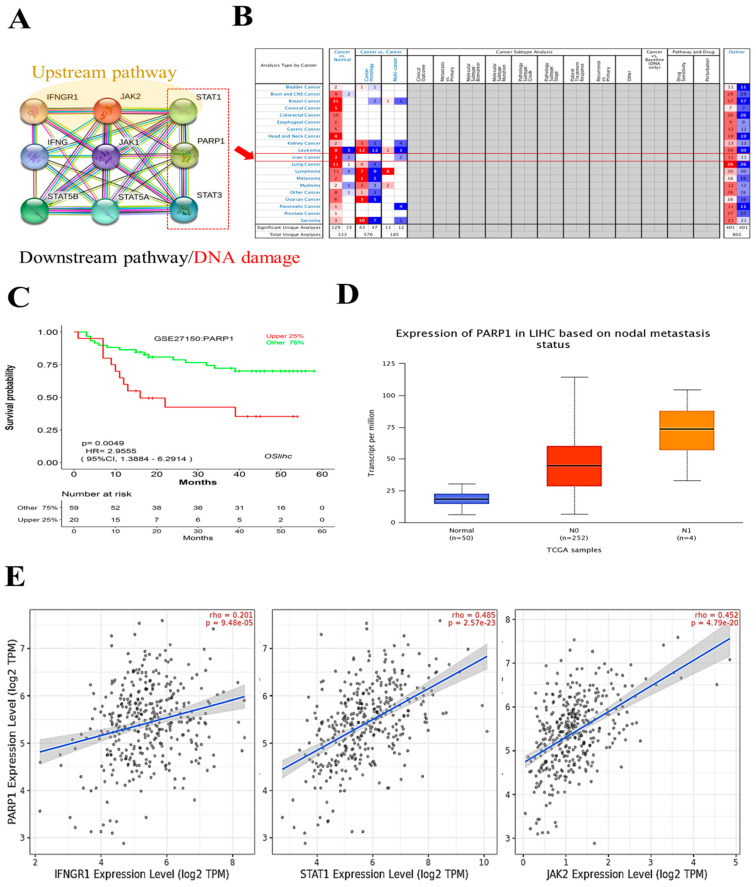
PARP1 expression associated with poor liver cancer patients’ overall survival. (**A**) Schematic representation of the interaction of the IFNGR-JAK-STAT1-PARP1 signaling proteins with the STRING analysis. (**B**) Transcription level of PAPR1 in different types of cancers (oncomine and TCGA) types. (**C**) K-M plot denoting the overall survival of liver cancer patients. (**D**,**E**) Expression and correlation analysis of PARP1 in liver cancer.

**Figure 7 cancers-13-02755-f007:**
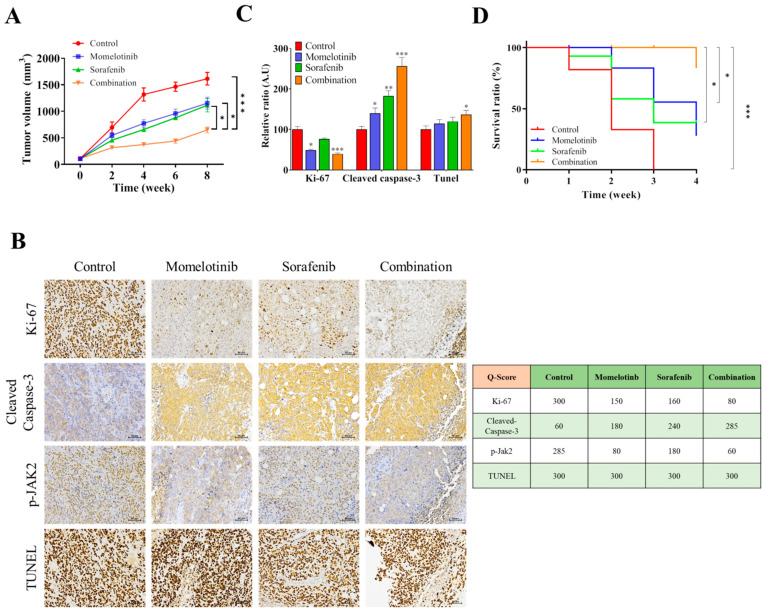
Momelotinib and sorafenib, in combination, potentiate the anti-proliferative effect of the single drug treatments in vivo. (**A**) The tumor volume over the time curve, the reduction in tumor burden was seen in combination treatment, as compared to the individual treatment groups. (**B**) Representative hematoxylin and eosin (H&E) stained images are shown, and IHC detected the expression of Ki-67 and cleaved caspase-3, p-Jak2 as the expression of activated Jak2 in the tumors. in the tumors. In addition, apoptotic cells in the tumors from mice were detected by the TUNEL assay. (**C**) The data were quantified and are represented as the means ± SD from 3 independent experiments. Combination vs. Momelotinib *** *p* < 0.01; combination vs. sorafenib * *p* < 0.05 and ** *p* < 0.01. Scale bar: 100 μm. (**D**) The overall survival rate of Xenograft mice.

**Figure 8 cancers-13-02755-f008:**
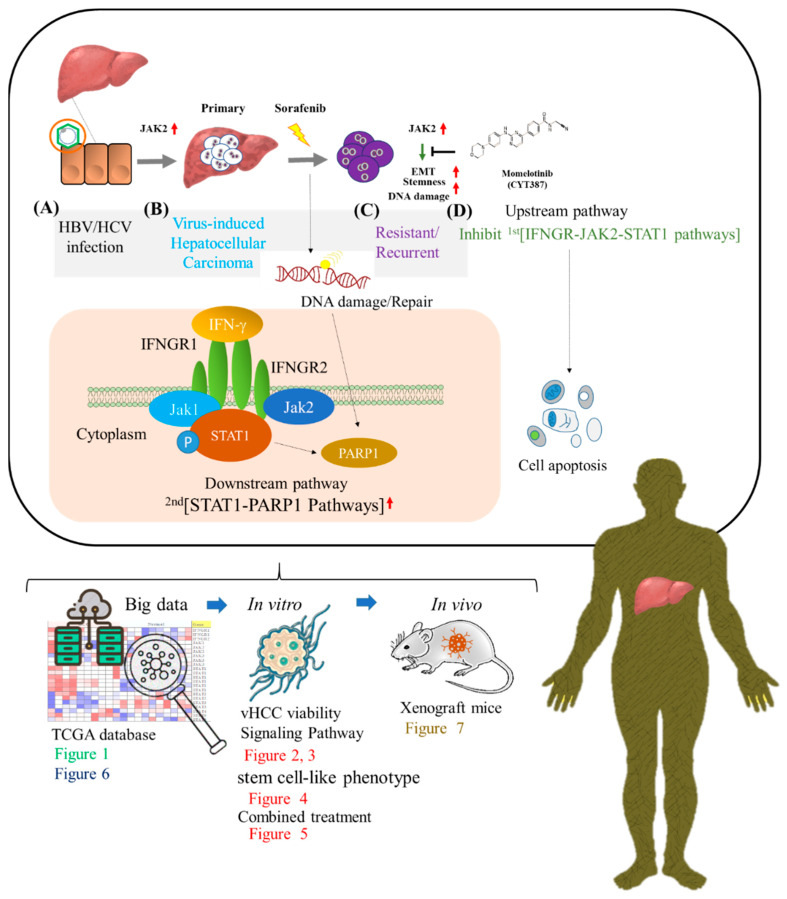
Schematic diagram and experimental designs. Jak2 overexpression afforded drug resistance to cancer cells, specifically in HBV/HCV-infected HCC, and the inhibition of Jak2 by momelotinib (JAK2 inhibitor) effectively suppressed HCC. This cancer development process includes four stages: (**A**) HBV/HCV infection. (**B**) Virus-induced hepatocellular carcinoma. (**C**) Sorafenib treatment causes DNA damage and initiates the STAT1-PARP1 pathway for DNA repair, which then forms resistant/recurrent cancer cells. (**D**) Momelotinib inhibits the upstream pathway of IFNGR-JAK2-STAT1, inhibits the downstream STAT1-PARP1 DNA repair pathway, and causes cancer cell apoptosis.

## Data Availability

The datasets used and analyzed in the current study are publicly-accessible as indicated in the manuscript.

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
