# Peer review of "Induced Mitochondrial Alteration and DNA Damage via IFNGR-JAK2-STAT1-PARP1 Pathway Facilitates Viral Hepatitis Associated Hepatocellular Carcinoma Aggressiveness and Stemness"

_cancers, 2021, doi:10.3390/cancers13112755_

Round 1

Reviewer 1 Report

In this manuscript, Cherng et al explore the role of IFNGR-JAK2-STAT1-PARP1 pathway in viral HCC.

They use nude mouse, patients’ samples and cell in culture to address their hypothesis. In this study author used two different drugs namely Momelotinib and Sorafenib, and they showed that both drugs significantly supressed the proliferation and effectively reduce the tumor burden. In addition author use gene expression omnibus dataset to address their hypothesis. There is few interesting observation, however, some concern need to be further clarification. In addition, the title mentioned the role of IFNGR-JAK2-STAT1-PARP1 pathway in VHCC but in this manuscript I have not find no such correlation in pathway.

My Comments:

  1. Author suggested momelotinib treatment inhibitis JAK2 protein and gene expression. But, I have not found any change in protein and mRNA expression under momelotinib treatment (FIG. 2E). The phosphorylation of JAK2 is inhibited. Author should clarify or make necessary correction.
  2. What is the p-JAK2 status in FIG.2B. Please mention the THLE-2 and HepG2 cell line in materials and methods section.
  3. What is the protein expression profile of IFNGR under the drugs?
  4. How author correlated the IFNGR-JAK2-STAT1-PARP1 pathway in viral HCC? Please clarify.
  5. The Fig.8. is not clear at all. Please provide a clear schematic diagram.
  6. Fig.4. Please make necessary correction in figure legend. ***p<0.001 is not in any panel in fig.4.
  7. All the figures are very low in quality. It is very difficult to read and understand the fig. Please provide high resolution fig. In addition there is no supplementary table.
  8. There are few capital letters in between two wards. Author should check the English and type error.

Author Response

We accordingly response the questions raised by the Reviewer as follows:

Response to Reviewers:

Point-by-point responses to reviewer’s comments – Reviewer 1:

Q1: Author suggested momelotinib treatment inhibitis JAK2 protein and gene expression. But, I have not found any change in protein and mRNA expression under momelotinib treatment (FIG. 2E). The phosphorylation of JAK2 is inhibited. Author should clarify or make necessary correction.

A1: We thank the reviewer for this comments. Please see the Discussion section in the revised manuscript.

Line 566:

Discussion:

“…Previous research indicated that overall Jak2 expression is related to the overall sur-vival in HCC patients [25]. Thus, the Jak family gene might be playing a key role in pa-tients with HCC. Verstovsek et al. [26] demonstrated the efficacy of momelotinib as a potent inhibitor of Jak1 and Jak2 in patients with primary and secondary myelofi-brosis. The inhibitory effect of Momelotinib (CYT387) on JAK2 phosphorylation is con-sistent with previous reports [26]. Momelotinib (CYT387) is an ATP-competitive small molecule that potently inhibits JAK1/JAK2 kinases. Furthermore, we found that momelotinib significantly inhibited the growth of HCC cells (Figure 2) and inhibit the phosphorylation of Jak2 and STAT3 and reduce the expression of p-Jak2, which veri-fied the importance of momelotinib in targeting Jak2 and reducing tumorigenesis in HCC.”

Q2: What is the p-JAK2 status in FIG.2B. Please mention the THLE-2 and HepG2 cell line in materials and methods section.

A2: We thank the reviewer for these comments. In the revised manuscript, we have added the p-JAK2 status in Fig. 2B. We have mentioned the THLE-2 and HepG2 cell line in materials and methods section.

Please see line 373:

“3.2. Momelotinib Inhibits Jak2 Expression, leading to Decreased Cell Viability

We investigated the potential of the Jak inhibitor momelotinib (CYT387; Figure 2A) to inhibit Jak2. As expected, the expression of Jak2 and p-Jak2 in vHCC cell lines SNU-475 and SNU-387 was higher than that in nvHCC (HepG2) and normal liver (THLE-2) cell lines (Figure 2B). Then, using the CompuSyn software to calculate the IC50 values at 48h for all four cell lines are as follows: SNU-475=6.3236, SNU-387=5.59423, HepG2= 6.89772, and THLE-2=12.4946. According to this result, we set 5mM as the subsequent experimental condition of the SNU cell line. The cell prolif-eration, as well as Jak2 luciferase activity of both SNU-475 and SNU-387 cell lines, sig-nificantly decreased under treatment with 5mM momelotinib in a dose-dependent manner (Figure 2C and 2D). We also confirmed the protein expression profile of IFNGR/JAK2 axis under the drugs. Momelotinib inhibits the phosphorylation of Jak2 and STAT3, and the inhibitory effect of this phosphorylation may affect the subsequent molecular signal transmission of liver cancer cells (Figure 2E). These results suggest that vHCC is sensitive to the inhibition of P-Jak2 by momelotinib..”

Please see line 128:

“2.2. Cell Lines and Reagents

The human SNU-387 and SNU-475 HCC cell lines were purchased from American Tissue Culture Collection (ATCC). The cells were maintained under conditions rec-ommended by the vendor, cultured in Roswell Park Memorial Institute (RPMI) 1640 Medium (Thermo Fisher Scientific Inc, Waltham, MA, USA.) supplemented with 10% fetal bo-vine serum (FBS) and 1% penicillin-streptomycin (Invitrogen, Life Technolo-gies) at 37°C, in a 5% humidified CO2 incubator. THLE-2 cell line was derived from primary normal liver cells by infection with SV40 large T antigen, HepG2 was derived from a liver hepatocellular carcinoma both were purchased from ATCC.THLE-2 and HepG2 cell lines were maintained in DMEM medium (Dulbecco's Modified Eagle Me-dium; Gibco, Invitrogen), and all cells were incubated at 37℃ in a humidified incuba-tor containing 5% CO2 in air. Cells were sub-cultured at 80%–90% conflu-ency. The Jak inhibitor momelotinib (CYT387) [21] and sorafenib (Catalog No. S7397), purchased from Selleck Chemicals, were dissolved in DMSO.”

Q3: What is the protein expression profile of IFNGR under the drugs?

A3: We thank the reviewer for these comments. We confirmed the protein expression profile of IFNGR under the drugs in Fig. 2E.

Please see line 373:

“3.2. Momelotinib Inhibits Jak2 Expression, leading to Decreased Cell Viability

We investigated the potential of the Jak inhibitor momelotinib (CYT387; Figure 2A) to inhibit Jak2. As expected, the expression of Jak2 and p-Jak2 in vHCC cell lines SNU-475 and SNU-387 was higher than that in nvHCC (HepG2) and normal liver (THLE-2) cell lines (Figure 2B). Then, using the CompuSyn software to calculate the IC50 values at 48h for all four cell lines are as follows: SNU-475=6.3236, SNU-387=5.59423, HepG2= 6.89772, and THLE-2=12.4946. According to this result, we set 5mM as the subsequent experimental condition of the SNU cell line. The cell prolif-eration, as well as Jak2 luciferase activity of both SNU-475 and SNU-387 cell lines, sig-nificantly decreased under treatment with 5mM momelotinib in a dose-dependent manner (Figure 2C and 2D). We also confirmed the protein expression profile of IFNGR/JAK2 axis under the drugs. Momelotinib inhibits the phosphorylation of Jak2 and STAT3, and the inhibitory effect of this phosphorylation may affect the subsequent molecular signal transmission of liver cancer cells (Figure 2E). These results suggest that vHCC is sensitive to the inhibition of P-Jak2 by momelotinib.”

Q4: How author correlated the IFNGR-JAK2-STAT1-PARP1 pathway in viral HCC? Please clarify.

A4: We thank the reviewer for this comment. In the revised manuscript, we added the correlation analysis of IFNGR-JAK2-STAT1-PARP1 to connect the discovery of this novel pathway.

Please see line 306:

“3. Results

3.1. IFNGR-JAK-STAT1 Signaling Pathway Is Activated in vHCC

We investigated the expression of IFNGR-JAK-STAT in hepatitis C virus (HCV)-associated HCC (vHCC), non-HCV-associated HCC (nvHCC), and normal liver samples. The robust microarray analysis expression data for liver samples from GEO datasets GSE14323 were explored, and 10 patients containing both normal liver and vHCC sample pairs were obtained after gene expression profiling within this sample group by using a heatmap. Interestingly, the expression of Jak2 in HCC cells was high-er than that in normal liver cells (Figure 1A). The differential expression of IFNGR1, IFNGR2, Jak1, Jak2, and STAT from the entire series was analyzed; as expected, the normal cohort (n = 19) showed lower Jak2 mRNA expression than did the HCC (HCV-infected) cohort (n = 38), and the GSE121248 cohort showed the similar result that Jak1 mRNA expression is higher in HCC (HBV-infected, n = 70) than in the nor-mal liver (n = 37; Figure 1B). Protein-protein interaction (PPI) network of IFNGR-JAK-STAT signaling proteins by STRING, clustering into 3 groups with blue, green, and red, representing (Figure 1C). Intriguingly, the expression of Jak2 in nvHCC groups from TCGA LIHC data remained similar compared with the normal liver group (Figure 1D), suggesting that Jak2 appears to be overexpressed only in vHCC. Figure 1E summarizes the heat map of different Hepatitis Virus-Associated Liver can-cer samples and shows the connection between the JAK2, STAT1 expression, and virus infection. From the results in Figure 1E, we analyzed the overall survival of HCC pa-tients with viral and non-viral infections. First, patients with vHCC have a poor over-all survival rate. When the experiment distinguished between vHCC and HCC pa-tients, we further discovered a higher expression of STAT1 and Jak2 is also associated with lower overall survival of patients with vHCC (Figure 1F). Further analyzing the relationship between the STAT1 gene expression and the overall survival of patients. We were surprised to find that the patients with high STAT1 expression have a lower overall survival rate in the vHCC group and it is statistically significant. In addition, this phenomenon was not found in the overall survival of HCC patients. Figure 1G shows the high correlation between JAK2, and STAT1 in TCGA (gene expression) and SHH vHCC patients (protein expression) cohort. In addition, we also performed an association analysis of gene expression (Supplementary Figure S1), and the results showed that the expression of JAK2 was significantly associated with IFNGR1/2 and STAT1 (P <0.001). The expression of PARP1 was significantly associated with STAT1 (P <0.001). The STAT1 is a downstream gene of JAK2, a novel pathway of IFNGR-JAK2-STAT1-PARP1 is connected in series.”

Q5: The Fig.8. is not clear at all. Please provide a clear schematic diagram.

A5: We thank the reviewer for these comments. We add a clearer mechanism description.

Please refer to the revised Figure 8

Please see the line 649:

“Figure 8. Schematic diagram and experimental designs. Jak2 overexpression afforded drug resistance to cancer cells, specifically in HBV/HCV-infected HCC, and the inhibition of Jak2 by momelotinib (JAK2 inhibitor) effectively suppressed HCC.This cancer development process includes four stages: (A) HBV/HCV infection. (B) Virus-induced hepatocellular carcinoma. (C) Sorafenib treatment causes DNA damage and initiates the STAT1-PARP1 pathway for DNA repair, which then forms cancer cells Resistant/ Recurrent. (D) Momelotinib inhibits the upstream pathway of IFNGR-JAK2-STAT1, inhibits the downstream STAT1-PARP1 DNA repair pathway, and causes cancer cell apoptosis”

Q6: Fig.4. Please make necessary correction in figure legend. ***p<0.001 is not in any panel in fig.4.

A6: We thank the reviewer for these comments. In the revised manuscript, we have corrected the legend of Figure 4.

Please see the line 428:

“Figure 4. Momelotinib suppresses the stem cell-like phenotype of vHCC. (A) Tumorsphere-forming ability of SNU-475 and SNU-387 cells and graphical quantitation compared with their untreated control counterparts. (B) The colo-ny-forming ability of SNU-475 and SNU-387 cells and quantitation compared with their treated and untreated control counterparts. (C) Protein and mRNA expression of stemness markers under treatment with momelotinib. β-actin was used as a loading control. (D) ALDEFLUOR assay showed that momelotinib treatment prominently reduces the activity of ALDH1 in SNU-475 and SNU-387 cells in a dose-dependent manner. The Momelotinib treatment dose and time with 5mM and 48h. * p < 0.05; ** p < 0.01. Scale bar: 100 μm.”

Q7: All the figures are very low in quality. It is very difficult to read and understand the fig. Please provide high resolution fig. In addition there is no supplementary table.

A7: We thank the reviewer for this comments. In the revised manuscript, we have improved the resolution of the image and added related supplementary table.

Q8: There are few capital letters in between two wards. Author should check the English and type error.

A8: We thank the reviewer for this comments. In the revised manuscript, we have check the English and type error.

Reviewer 2 Report

I have the following suggestions for the authors:

  • Authors can calculate a synergy score for combined Momelotinib and Sorafenib treatment using synergy estimation tools like SynergyFinder(https://academic.oup.com/nar/article/48/W1/W488/5815821)
  • Figure 8. is confusion. Authors can describe what does A,B C and D in the figure
  • Figure 1 (Panel I): Authors can calculate a p-value for Q-score comparing samples in different groups
  • Authors should explain that why they choose to use Momelotinib which is both (JAK1 and 2) inhibitor instead of JAK2 specific inhibitors (eg Fedratinib) or other JAK2 inhibitors ( e.g. Gandotinib, BMS-911543, XL019).
  • What was the number of mice used for experimentation in Figure 7 (D)
  • Figures have poor visibility please improve it.

Author Response

Point-by-point responses to reviewer’s comments – Reviewer 2:

Q1: Authors can calculate a synergy score for combined Momelotinib and Sorafenib treatment using synergy estimation tools like SynergyFinder(https://academic.oup.com/nar/article/48/W1/W488/5815821)

A1: We thank the reviewer for this comments. In the revised manuscript, we have calculated a synergy score for combined Momelotinib and Sorafenib treatment using synergy estimation tools.

Please see line 435:

“3.5. Momelotinib Treatment Increased Sorafenib Sensitivity of vHCC Cells

Momelotinib is a potent ATP-competitive inhibitor of JAK2, it has been used for targeted therapies of myeloproliferative tumors. A previous study showed that treat-ment with sorafenib significantly improved the survival of patients with solid tumors (Abou-Alfa et al. 2006). In our results, the combination of momelotinib and sorafenib suppressed the proliferation of SNU-387 cells (Figure 5A). In order to inhibit the effect of JAK2 pathway activation on drug resistance, we tested whether the combination of momelotinib and sorafenib can remarkably suppress vHCC cell proliferation and col-ony formation. We observed that the combination treatment with momelotinib and sorafenib synergistically inhibited vHCC cell proliferation and colony formation by inducing apoptosis. Figure 5B shows the momelotinib-sorafenib combination testing on SNU-387 and the effect of treatment on colony-forming capacity of SNU-387 cells and treated with the drugs (sorafenib and momelotinib in combination) at the 2.5mM and 5mM concentration for 48 hours. We also calculate a synergy score for combined Momelotinib and Sorafenib treatment using synergy estimation tools (SynergyFinder, https://academic.oup.com/nar/article/48/W1/W488/5815821) as shown in supplemen-tary Figure S2. SynergyFinder extracts the data for all the possible drug combination pairs from the data and visualize the synergy map calculated from synergy scoring models. The result includes heatmap, contour plot and interactive 3D surface which was demonstrated to prioritize synergistic drug pairs with higher efficacy and lower toxicity as top hits, providing thus an increased likelihood for their clinical success. In addition, we also analyzed the drug sensitivity of the HCC (HepG2) cell line, shown in Supplementary Figure S3 and S4. The results indicated that HepG2 cells had less obvious impacts on JAK2 inhibitor (Momelotinib) compared with SNU-387. Momelotinib is a selective inhibitor of JAK1 and JAK2, we speculated that the hepa-toma cell lines (SNU-387) infected by the virus might induce JAK2 expression through the IFN gamma pathway. This data might be supporting our proposed model of JAK2 expression in vHCC. According to these results, we found the optimized parameter of combination. In Figure 5C, we did the apoptosis analysis by flow cytometer for An-nexin-V+ and 7-AAD stained cells. The results indicated the elevated apoptosis in the combo treatment. Determining the expression of apoptosis marker is important to un-derstand the functions of molecular mechanism. Hence, Figure 5D showed the repre-sentative Western blot images of the apoptosis markers in SNU-387. The expression levels of apoptosis markers including p-JaK2 and Bcl-x1 were decline, the cleaved-PARP, cleaved-Caspase 7, and 9 were the rise in the combination-treated group. This result shows the therapeutic potential of oncogenic/Stemness JAK2 path-way inhibitors on viral hepatitis. As stated above, these results demonstrated the po-tential inhibition of vHCC cells by treatment with momelotinib–sorafenib combina-tion.”

Q2: Figure 8. is confusion. Authors can describe what does A,B C and D in the figure

A2: We thank the reviewer for this comments. In the revised manuscript, we have described what does A, B C and D in the figure.

Please see line 649:

“Figure 8. Schematic diagram and experimental designs. Jak2 overexpression afforded drug resistance to cancer cells, specifically in HBV/HCV-infected HCC, and the inhibition of Jak2 by momelotinib (JAK2 inhibitor) effectively suppressed HCC.This cancer development process includes four stages: (A) HBV/HCV infection. (B) Virus-induced hepatocellular carcinoma. (C) Sorafenib treatment causes DNA damage and initiates the STAT1-PARP1 pathway for DNA repair, which then forms cancer cells Resistant/ Recurrent. (D) Momelotinib inhibits the upstream pathway of IFNGR-JAK2-STAT1, inhibits the downstream STAT1-PARP1 DNA repair pathway, and causes cancer cell apoptosis”

Q3: Figure 1 (Panel I): Authors can calculate a p-value for Q-score comparing samples in different groups

A3: We thank the reviewer for this comments. In the revised manuscript, we have calculated a p-value for Q-score comparing samples in different groups.

Please see Figure 1 (I).

Q4: Authors should explain that why they choose to use Momelotinib which is both (JAK1 and 2) inhibitor instead of JAK2 specific inhibitors (eg Fedratinib) or other JAK2 inhibitors ( e.g. Gandotinib, BMS-911543, XL019).

A4: We thank the reviewer for this comments. In the revised manuscript, we have explained that why they choose to use Momelotinib which is both (JAK1 and 2) inhibitor instead of JAK2 specific inhibitors or other JAK2 inhibitors.

Please see line 528:

Discussion:

“The Janus kinase signal transducer and activator of transcription (JAK-STAT) signal is essential for a variety of cellular processes including survival, differentiation and proliferation. Currently, 4 therapeutic JAK2 inhibitors (ruxatinib, melatinib, momolotinib and pacomatinib) have been approved or are in advanced clinical studies. Molotinib (CYT387) is a dual inhibitor of JAK1 and JAK2. The related downstream JAK1 and JAK2 in the IFNGR pathway induced by viral infection are related. Hence, we selected this dual inhibitor to evaluate its therapeutic effect. In addition, this is the first study to explore the application of Molotinib to viral liver cancer. In the present study, we first demonstrated that the expression of Jak1/2 and PAPR1 is significantly upregulated in vHCC than in nvHCC/normal liver tissues (Figure 1 and 6). In addition, the proteins of IFN gamma-related pathways are activated after virus infection (Fig-ure 1B). This result suggests the difference in signal transmission between viral liver cancer and general liver cancer. Based on this observation, we are linked to the rele-vant targets of the JAK family and the potential applications of targeted therapy in-hibitors. To further confirm our hypothesis that the JAK family and virus infection cause cancer. We also analyzed the overall survival between vHCC and HCC. Among them, STAT1 expression in both is a significant difference. In virus infection, the per-formance of IFN-gamma is an important indicator. IFN-gamma binds to nearby unin-fected cell membrane receptors, stimulating signaling pathways to interfere with virus replication; stimulating cells to hydrolyze pathogenic proteins to prevent cells from being infected by the same or different viruses. In addition, STAT1 is a downstream gene of JAK2 in IFN-gamma signaling pathways, which can infer the significance of JAK2 in viral infection. Previous research indicated that overall Jak2 expression is re-lated to the overall sur-vival in HCC patients [30]. Thus, the Jak family gene might be playing a key role in pa-tients with HCC. Verstovsek et al. [31] demonstrated the effi-cacy of momelotinib as a potent inhibitor of Jak1 and Jak2 in patients with primary and secondary myelofibrosis. The inhibitory effect of Momelotinib (CYT387) on JAK2 phosphorylation is consistent with previous reports [32]. Momelotinib (CYT387) is an ATP-competitive small molecule that potently inhibits JAK1/JAK2 kinases. Further-more, we found that momelotinib significantly inhibited the growth of HCC cells (Fig-ure 2) and inhibit the phosphorylation of Jak2 and STAT3 and reduce the expression of p-Jak2, which verified the importance of momelotinib in targeting Jak2 and reducing tumorigenesis in HCC. Often HCC patients show sorafenib therapy resistance [33]. Drug resistance is a major challenge in anticancer therapy. CSCs provide an alterna-tive explanation for the aforementioned therapeutic challenges of several cancers [34]. This small population of cancer cells has stem cell-like features such as tumorigenicity, self-renewal, and more resistance to chemotherapeutic agents than that shown by cancer cells [35, 36]. Treatment of vHCC with momelotinib reduced the expression of cancer stemness markers, such as CD133, KLF4, and SOX2, and decreased ALDH1 ac-tivity (Figure 4), suppressing tumorsphere and colony formation of HCC cells.”

Q5: What was the number of mice used for experimentation in Figure 7 (D)

A4: We thank the reviewer for this comments. In the revised manuscript,we added the number of mice used for experimentation in Figure 7 (D).

Please see line 275:

“2.16. Animal Studies

All animal experiments and maintenance were in strict compliance with the An-imal Use Protocol Taipei Medical University (protocol LAC-2014-0170). The an-ti-proliferative effect of momelotinib and sorafenib in combination with vHCC cells in vivo investigated, athymic nude mouse models bearing HCC cell xenografts were es-tablished. Five-week-old male athymic nude mice were used for this study. The animal experiment is set to six mice per group. The mice were maintained under patho-gen-free conditions and were provided with sterilized food and water. First, 1 × 106 SNU-387 cells were subcutaneously injected into the right flank near the hind leg of each nude mouse. When the mice had palpable tumors (tumor volume of approxi-mately 100 mm3), they were randomly divided into control (100 µL of normal saline [NS] by intraperitoneal injection plus 100 µL of 1% DMSO) and 0.5% carboxymethyl cellulose ([CMC]-Na sterile water), momelotinib (200 mg/kg/day by intraperitoneal in-jection plus 100 µL of 1% DMSO and 0.5% CMC-Na sterile water), sorafenib (30 mg/kg/day by intragastric administration plus 100 µL NS by intraperitoneal injection), and combination (momelotinib, 30 mg/kg/day by intraperitoneal injection plus soraf-enib 30 mg/kg/day by intragastric administration) groups (n = 6 animals/group). The treatments were performed 5 times/week for 4 weeks. The tumor volume was detected every week and was calculated using the following formula: volume (V) = π/6 × length × width × height. After 4 weeks, mice were humanely euthanized, and the tumors were isolated for further analyses.”

Q6: Figures have poor visibility please improve it.

A6: We thank the reviewer for these comments. In the revised manuscript, we have improved the resolution of the Figure.

Round 2

Reviewer 1 Report

I have no more comments.